# Genomic landscape of extraordinary responses in metastatic breast cancer

Sun Min Lim[1,8], Eunyoung Kim [2,8], Kyung Hae Jung[3,8], Sora Kim[2], Ja Seung Koo[4], Seung Il Kim[5], Seho Park[5], Hyung Seok Park[5], Byoung Woo Park[5], Young Up Cho [5], Ji Ye Kim[5], Soonmyung Paik [6], Nak-Jung Kwon[7], Gun Min Kim[1], Ji Hyoung Kim [1], Min Hwan Kim[1], Min Kyung Jeon[1], Sangwoo Kim [2✉] & Joohyuk Sohn [1✉]

Extreme responders to anticancer therapy are rare among advanced breast cancer patients. Researchers, however, have yet to investigate treatment responses therein on the whole exome level. We performed whole exome analysis to characterize the genomic landscape of extreme responders among metastatic breast cancer patients. Clinical samples were obtained from breast cancer patients who showed exceptional responses to anti-HER2 therapy or hormonal therapy and from those who did not. Matched breast tumor tissue (somatic DNA) and blood samples (germline DNA) were collected from a total of 30 responders and 15 non-responders. Whole exome sequencing using Illumina HiSeq2500 was performed for all 45 patients (90 samples). Somatic single nucleotide variants (SNVs), indels, and copy number variants (CNVs) were identified for the genomes of each patient. Group-specific somatic variants and mutational burden were statistically analyzed. Sequencing of cancer exomes for all patients revealed 1839 somatic SNVs (1661 missense, 120 nonsense, 43 splice-site, 15 start/stop-lost) and 368 insertions/deletions (273 frameshift, 95 in-frame), with a median of 0.7 mutations per megabase (range, 0.08 to 4.2 mutations per megabase). Responders harbored a significantly lower nonsynonymous mutational burden (median, 26 vs. 59, $P = 0.02$) and fewer CNVs (median 13.6 vs. 97.7, $P = 0.05$) than non-responders. Multivariate analyses of factors influencing progression-free survival showed that a high mutational burden and visceral metastases were significantly related with disease progression. Extreme responders to treatment for metastatic breast cancer are characterized by fewer non-synonymous mutations and CNVs.

[1] Division of Medical Oncology, Department of Internal Medicine, Yonsei Cancer Center, Yonsei University College of Medicine, Seoul, Korea. [2] Department of Biomedical Systems Informatics, Brain Korea 21 PLUS Project for Medical Sciences, Yonsei University College of Medicine, Seoul, Korea. [3] Department of Oncology, Asan Medical Center, University of Ulsan College of Medicine, Seoul, Korea. [4] Department of Pathology, Yonsei University College of Medicine, Seoul, Korea. [5] Department of General Surgery, Yonsei University College of Medicine, Seoul, Korea. [6] Severance Biomedical Science Institute and Department of Medical Oncology, Yonsei University College of Medicine, Seoul, Korea. [7] Macrogen, Inc, Seoul, Korea. [8] These authors contributed equally: Sun Min Lim, Eunyoung Kim, Kyung Hae Jung. ✉email: swkim@yuhs.ac; ONCOSOHN@yuhs.ac

Breast cancer is a heterogeneous disease categorized according to three therapeutic biomarkers: (1) estrogen receptor (ER)-positive cancer, (2) HER2 (ERBB2)-amplified cancer, and (3) triple-negative cancer. ER-positive cancers can be further separated into those with higher proliferative (luminal A-like) and lower proliferative (luminal B-like) capabilities. Nevertheless, even with the predictive markers, patients still exhibit various responses to their designated therapies: for example, almost 50% of all ER-positive patients with advanced disease do not respond to endocrine therapy, and a complete response is rarely encountered. As such, interest in the causes that underly these varying responses is growing.

Cancer acquires successive genetic alterations, such as point mutations and copy number changes, during clonal evolution. Breast cancer is characterized by genomic instability that drives tumor heterogeneity, including mutations, copy number alterations, and chromosomal structural rearrangements[1]. Recent molecular advances have helped outline the genomic landscape of breast cancer, and multiple mutational signatures have been suggested[2]. However, these studies do not contain information on treatment responses.

Exceptional responses, such as complete responses or durable partial responses, are rare in solid tumors. However, case studies of extraordinary responses to targeted therapeutics have been reported, suggesting that some somatic alterations in a patient's tumor may elicit extreme responses[3–5]. In the cancer genome, the prevalence of somatic mutations can vary greatly between cancers, ranging from about 0.001 per megabase (Mb) to more than 400 per Mb[6]. Notably, potentially curable cancers, such as hematologic malignancies, and childhood cancers, such as pilocytic astrocytoma and acute lymphoblastic leukemia, carry the fewest mutations, while chemotherapy-resistant cancers, such as malignant melanoma, show the opposite, with high mutational burden. Accordingly, we hypothesized that cancers with fewer mutations may be more sensitive to anticancer treatment, owing to simplicity of their genome. In line with this, we also hypothesized that breast cancer patients who show extreme sensitivity to anticancer treatment may harbor fewer genomic alterations.

In this study, we prospectively collected primary tumor tissue and blood samples from extreme responders to anti-ER or anti-HER2 therapy and non-responders, and performed whole exome analysis of these patients. The primary aim of this study was to compare mutational burden and copy number variations between the extreme responders and non-responders.

## Results

**Design of the study**. The overall design of the study is depicted in Fig. 1. A total of 2548 cases of metastatic breast cancer were searched for extreme responders and non-responders. Among 89 patients who met the inclusion criteria, we were able to obtain informed consent consecutively from 71 patients (43 extreme- and 23 non-responders) who visited an outpatient clinic from April 2013 to February 2019. Out of 43 extreme responders, 13 were excluded due to insufficient primary breast tumor materials for whole exome sequencing. Likewise, six out of 23 non-responder patients were excluded. One other non-responder who expired before blood collection was also excluded. Consequently, a total of 30 responders and 15 non-responders were finally analyzed (Fig. 1).

The average depths of sequencing were 107 for tumor samples and 65 for blood samples. Based on the list of somatic alterations, group-specific variations and mutational burden were explored. Functional analysis of the group-specific variations, including frequently aberrated regions and genomic instability, was conducted to explain differences between the two groups. An overview of the genome variant analysis is presented in Supplementary Fig. 1.

**Patient characteristics**. The median age of all patients was 55 years (range, 36–72), and there were 38 ER-positive patients and 7 HER2-positive patients. All HER2-positive patients were ER-negative. The clinical characteristics of all patients are outlined in Table 1. According to initial stage, there were 7 (16%) patients with stage 1, 10 (22%) patients with stage 2, 8 (18%) patients with stage 3, and 20 (44%) patients with stage 4 disease. The most common site of metastasis or recurrence was the lungs (30%), followed by bone (27%), and liver (27%). Among the 30 responders, 24 were ER-positive (80%), and all showed a complete response or durable partial response to either aromatase inhibitors or tamoxifen. Among responders, 20 (67%) patients had received therapy as their first line of treatment, and 3 (10%) patients had experienced progression at the time of data cut-off (May 2019). Among non-responders, 10 (66%) patients had received therapy as their first line of treatment, and all patients had experienced progression at the data cut-off date. The median PFS of responders was not reached at the time of data cut-off, with a median follow up of 42 months, and the median PFS of non-responders was 5.5 months (95% CI, 3.86–7.14) (Fig. 2).

**Analysis of nonsynonymous mutational and CNV burden**. We compared mutational burden between the responder and non-responder groups with respect to the number of somatic SNVs (with indels) and somatic CNVs. Initially, a total of 13,617 somatic SNVs and 361 indels were called in the 45 patients. We further filtered out 11,778 SNVs that did not lead to protein alterations (mutations outside of exons and synonymous mutations). The remaining 1839 mutations included 1661 missense, 120 nonsense, 15 start-loss and stop-loss, 43 SNVs at a splice junction, and 368 indels (95 in-frame and 273 frameshift) (Fig. 3a). The overall nonsynonymous mutational burden was 0.99 (±0.95) per Mb (median, 0.72 per Mb), which is comparable to a previous report of 1.29 (±1.33) per Mb (median, 0.93 per Mb)[7]. On average, ER-positive patients had a mutational burden of 1.08 per Mb; HER2-positive patients had a mutation burden of 0.51 per Mb.

Next, we compared nonsynonymous mutational burdens between the responder and non-responder groups (Fig. 3b). The number of nonsynonymous mutations in the 30 responders ranged from 4 to 195 per patient, with an average of 41.47 and a median of 26 mutations. In contrast, the number in the 15 non-responders ranged from 10 to 210 per patient, with an average of 65.80 and a median of 59 mutations, an approximately two-fold increase. With only a few outlier patients, mutational burden could discriminate the two groups (Fig. 3a), and the difference in mutational burden was statistically significant (Wilcoxon rank-sum test, $P = 0.02$, one-side), confirming the hypothesis that extraordinary responders in metastatic breast cancer would have a lower tumor mutational burden than non-responders. Mutational burden in the ER-positive patients also showed a significant difference between the responders and non-responders (45.96 vs. 67.93, Wilcoxon rank-sum test, $P = 0.04$, one-side), with non-responders harboring more nonsynonymous mutations.

We further categorized all patients into two groups according to the median nonsynonymous mutational burden (median = 0.72 per Mb). In doing so, we sought to determine whether patients with lower nonsynonymous mutational burden would show longer PFS than patients with a higher nonsynonymous mutational burden. In survival analysis, we found that the median PFS of patients with the lower mutational burden was not reached, whereas the median PFS of patients with the higher

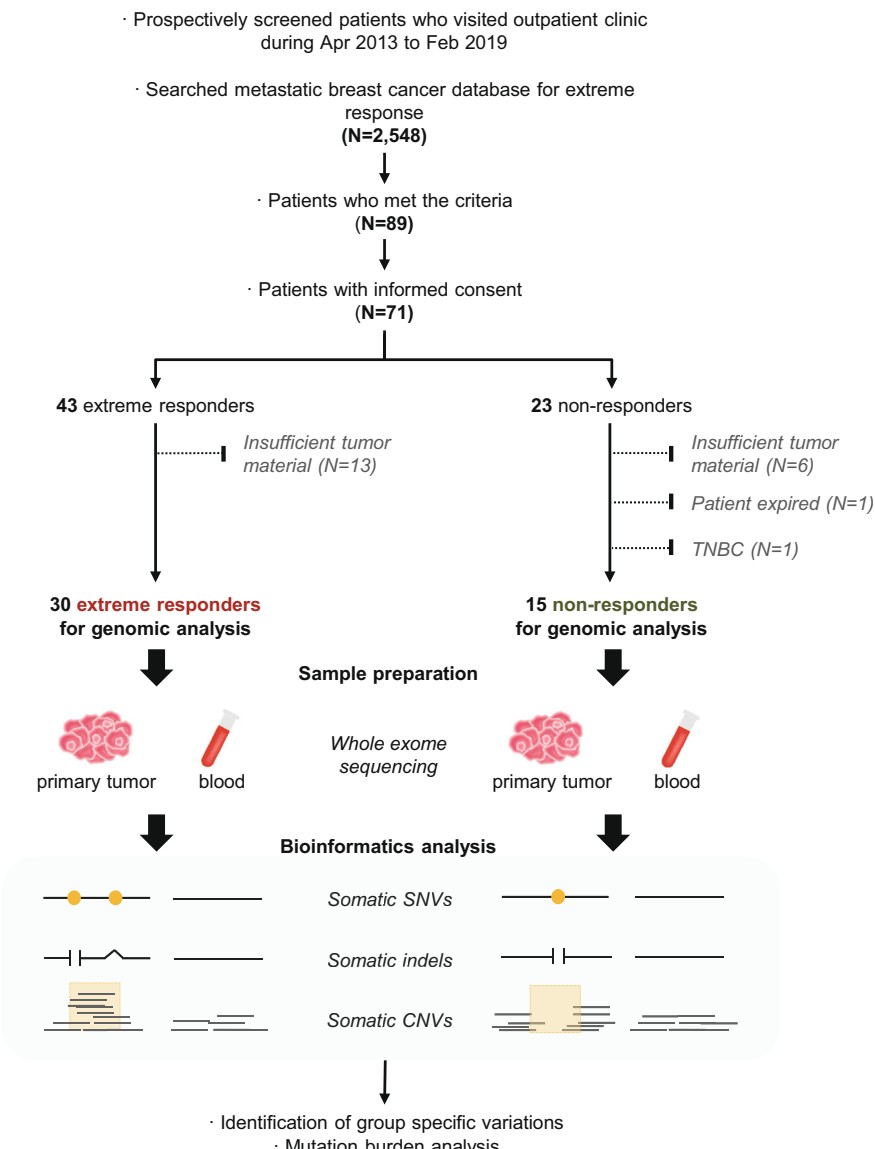

· Prospectively screened patients who visited outpatient clinic
during Apr 2013 to Feb 2019

· Searched metastatic breast cancer database for extreme
response
**(N=2,548)**

· Patients who met the criteria
**(N=89)**

· Patients with informed consent
**(N=71)**

**43 extreme responders**

*Insufficient tumor material (N=13)*

**23 non-responders**

*Insufficient tumor material (N=6)*

*Patient expired (N=1)*

*TNBC (N=1)*

**30 extreme responders**
**for genomic analysis**

**15 non-responders**
**for genomic analysis**

**Sample preparation**

*Whole exome sequencing*

primary tumor    blood

primary tumor    blood

**Bioinformatics analysis**

*Somatic SNVs*

*Somatic indels*

*Somatic CNVs*

· Identification of group specific variations
· Mutation burden analysis

**Fig. 1 Schematic diagram of the study protocol.** The overall study procedure is depicted to show patient enrollment and analysis.

nonsynonymous mutational burden was 5.9 months (95% CI, 3.61–8.18) (Supplementary Fig. 2a). Multivariate analyses of factors influencing PFS (mutational burden, initial metastasis, line of therapy, visceral metastasis) revealed that high mutational burden and visceral metastases were significantly related with disease progression (both $P < 0.05$) (Table 2).

Additionally, we compared CNV burden between the responders and non-responders. Due to ambiguity in the definition of CNV, CNV burden was measured using two different values: CNV total size (the size of genomic regions affected by CNVs) and CNV average size (CNV total size per the number of distinct CNVs) (Fig. 4a). The median CNV counts (the number of distinct CNVs) per patient were 13 (mean = 18.7) in responders and 12 (mean = 17.6) in non-responders (Wilcoxon rank-sum test $P = 0.45$, one-side). The median CNV gene counts (the number of genes with distinct CNVs) per patient were 134 (mean = 358.7) in responders and 594 (mean = 660.9) in non-responders (Wilcoxon rank-sum test $P = 0.05$, one-side). The median total lengths of CNVs per patient were 13.6 Mb (mean = 41.59 Mb) in responders and 97.7 Mb (mean = 77.85 Mb) in non-responders (Wilcoxon rank-sum test $P = 0.05$, one-side). The median average lengths of CNVs per patient were 0.99 Mb

(mean = 1.90 Mb) in responders and 3.94 Mb (mean = 5.99 Mb) in non-responders (Wilcoxon rank-sum test $P = 0.03$, one-side). We found that CNV burden was significant in discriminating between the two groups, expect for CNV count (W = 230.5, $P = 0.452$, one-side) and total length (W = 293, $P = 0.052$, one-side). Nevertheless, the total measured lengths of CNVs under both definitions in responders were consistently lower than those in non-responders: the total lengths of CNVs in the 15 non-responders ranged from 0.09 to 183 Mb, with a total sum of 1167 Mb, and those in the 30 responders ranged from 0 to 139 Mb, with a total sum of 1247 Mb (Fig. 4b). In addition, we categorized patients into two groups according to the median value of CNV burden (Supplementary Fig. 2b). In doing so, we found that the median PFS of patients with lower CNV burden was not reached, whereas the median PFS of patients with higher CNV burden was 34.4 months ($P = 0.032$).

**Analysis of group-specific variants and enrichment pathways.** Next, we investigated differences in genomic variants and enrichment pathways between the responders and non-responders (Supplementary Fig. 3). Fisher's exact test revealed

**Table 1 Clinical characteristics of all patients.**

| Pt | ER/HER2 | Age | Initial stage | Site of metastases | Treatment | PFS (mo.) |
|----|---------|-----|---------------|--------------------|-----------|-----------|
| 1 | HER2+ | 57 | 4 | Lung | Herceptin + taxane followed by herceptin maintenance | 86.0+ |
| 2 | HER2+ | 41 | 4 | Brain | TDM1 | 62.9+ |
| 3 | HER2+ | 49 | 4 | Bone, lung | Herceptin + taxane followed by herceptin maintenance | 49.9+ |
| 4 | HER2+ | 39 | 3 | Ovary | Herceptin + taxane followed by herceptin maintenance | 37.9 |
| 5 | HER2+ | 67 | 1 | Liver | Xeloda + lapatinib | 34.5 |
| 6 | HER2+ | 62 | 4 | Axillary lymph node, bone | Herceptin + taxane followed by herceptin maintenance | 34.4 |
| 7 | ER+ | 74 | 2 | Pleura | Letrozole | 100.3+ |
| 8 | ER+ | 58 | 4 | Ipsilateral axillary lymph node, lung, liver, bone, brain | Anastrozole | 100+ |
| 9 | ER+ | 56 | 1 | Lung | Letrozole | 87+ |
| 10 | ER+ | 49 | 3 | Lung | Tamoxifen | 73+ |
| 11 | ER+ | 48 | 4 | Pleura | Tamoxifen | 72.8+ |
| 12 | ER+ | 46 | 2 | Bone | Letrozole/leuprorelin | 71+ |
| 13 | ER+ | 61 | 3 | Pleura, mediastinal LN | Letrozole | 69+ |
| 14 | ER+ | 44 | 2 | Bone | Everolimus/exemestane | 63+ |
| 15 | ER+ | 60 | 2 | Liver | Letrozole | 62.1+ |
| 16 | ER+ | 54 | 4 | Liver | Letrozole | 62+ |
| 17 | ER+ | 52 | 2 | Pleura, lung | Anastrozole | 61.6+ |
| 18 | ER+ | 551 | 2 | Pleura, mediastinum | Letrozole | 61.4+ |
| 19 | ER+ | 691 | 1 | Pleura | Everolimus/exemestane | 61+ |
| 20 | ER+ | 512 | 2 | Lung | Everolimus/letrozole/leuprorelin | 61+ |
| 21 | ER+ | 592 | 3 | Bone | Letrozole | 61+ |
| 22 | ER+ | 58 | 1 | Lung | Arimidex | 60.5+ |
| 23 | ER+ | 50 | 3 | Multiple bone | Everolimus/letrozole/leuprorelin | 59+ |
| 24 | ER+ | 52 | 3 | Liver | Everolimus/letrozole/leuprorelin | 49+ |
| 25 | ER+ | 48 | 2 | Liver, ovary | Femara | 47.1+ |
| 26 | ER+ | 69 | 4 | Pericardial, ipsilateral cervical LN | Letrozole | 42+ |
| 27 | ER+ | 72 | 4 | Stomach | Letrozole | 34+ |
| 28 | ER+ | 51 | 2 | Lung | Letrozole/leuprorelin | 28+ |
| 29 | ER+ | 56 | 3 | Bone | Everolimus/exemestane | 23+ |
| 30 | ER+ | 52 | 4 | Bone | Paclitaxel | 10.6+ |
| *Non-responders* | | | | | | |
| 1 | HER2+ | 56 | 4 | Liver, lung | Herceptin + taxane | 7.2 |
| 2 | ER+ | 57 | 1 | Bone | Letrozole | 12.2 |
| 3 | ER+ | 59 | 4 | Bone | Letrozole, LY2835219 or placebo | 9.6 |
| 4 | ER+ | 40 | 4 | Lung | Letrozole | 8.1 |
| 5 | ER+ | 58 | 4 | Liver | Letrozole | 7.4 |
| 6 | ER+ | 49 | 4 | Bone, liver | Tamoxifen, Goserelin | 6.6 |
| 7 | ER+ | 56 | 4 | Bone, distant LN | Tamoxifen, Goserelin | 5.9 |
| 8 | ER+ | 55 | 4 | Lung | Letrozole | 5.5 |
| 9 | ER+ | 59 | 4 | Liver, axillary LN | Letrozole, palbociclib | 4.9 |
| 10 | ER+ | 49 | 4 | Brain, liver | Letrozole | 4.6 |
| 11 | ER+ | 43 | 1 | Lung | Letrozole | 4 |
| 12 | ER+ | 63 | 3 | Bone | Femara + ibrance | 2.3 |
| 13 | ER+ | 49 | 1 | Distant LN | Letrozole, palbociclib | 2.1 |
| 14 | ER+ | 36 | 2 | Bone | Tamoxifen | 1 |
| 15 | ER+ | 49 | 4 | Bone | Tamoxifen | 0.96 |

*LN* lymph node, *PFS* progression-free survival.

five non-responder-specific genes ($P < 0.05$): *AFF2, TTN, TP53, ATM,* and *MLLT4*. These genes have previously been shown to be related to biological pathways of cell cycle control, DNA damage repair, and apoptosis in breast cancer, either directly or indirectly[8]. In enrichment pathway analysis of mutated genes, alterations in genes important in ER signaling and ERBB2-related signal transduction were specific to non-responders (adjusted $P$ value $< 0.01$): Mutations in the three genes (*PIK3CA, AKT1,* and *ESR1*) associated with these two pathways have been shown to induce drug resistance in breast cancer[9–11].

*ESR1* gene mutations known to be associated with resistance to aromatase inhibitors in ER-positive metastatic breast cancer were found in two non-responders[12–14]. These mutations were located in known hotspot regions, Y537S and D538G, at the ligand-binding domain, which facilitates hormone-independent ER transcriptional activity. Mutations therein are known to lead to resistance to aromatase inhibitors and decreased sensitivity to tamoxifen and fulvestrant[11]. While there was one in-frame deletion located in the hinge domain in responders, the functional impact was unpredictable and might be tolerated.

Regarding CNVs, 1q (1q21.2, 1q32.3, 1q41, 1q44), 8q (8q11.22, 8q11.23, 8q12.3, 8q13.1, 8q13.2, 8q21.12, 8q24.13), and 17q25.2 were found to be specific to non-responders (Fisher's exact test $P < 0.05$). The gain of a whole long arm (1q) was a common

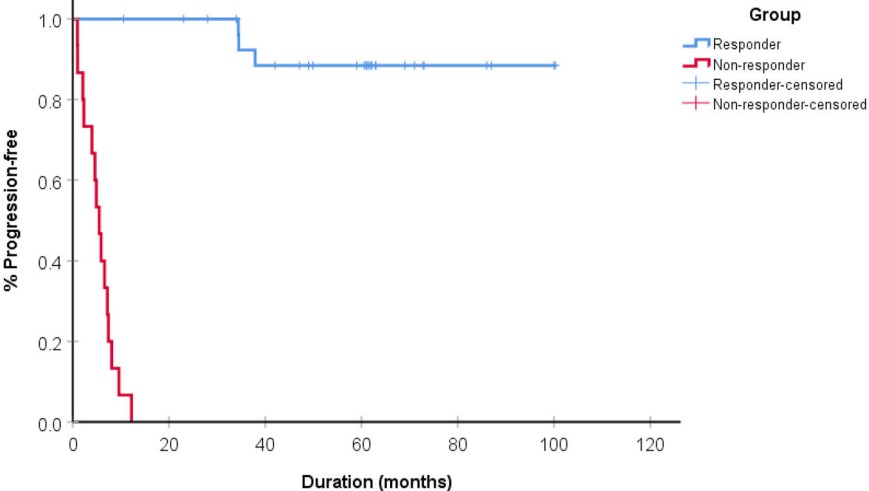

**Fig. 2 Kaplan–Meier curves comparing progression-free survival for responders and non-responders.** The median PFS of non-responders was 5.5 months (95% CI, 5.0–5.9), and the median PFS of responders was not reached (*P* < 0.001 by log-rank).

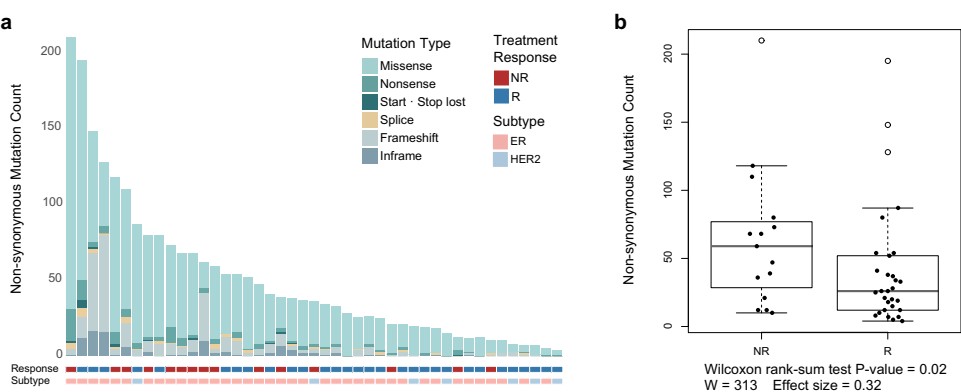

**Fig. 3 Exomic landscape of all patients and the comparison of non-synonymous mutation burden between responders and non-responders. a** Exomic landscape of all patients according to response, molecular subtype, and nonsynonymous mutations. The 13 samples represent the top 30 percent of the nonsynonymous mutational burden and consisted of eight samples from the 15 nonresponders and five samples from the 30 responders. **b** Comparison of nonsynonymous mutational burden between responders and non-responders. *P* values were calculated by the Wilcox rank-sum test (NR non-responder, R responder).

**Table 2 Univariate and multivariate analyses of parameters associated with progression.**

|  | Univariate analysis | | Multivariate analysis | |
| --- | --- | --- | --- | --- |
|  | Odds ratio | *P* | Odds ratio | *P* |
| Age |  |  |  |  |
| <55 | 1 |  |  |  |
| ≥55 | 1.1 | 0.851 |  |  |
| Mutational burden |  |  |  |  |
| Low | 1 |  | 1 |  |
| High | 3.563 | **0.028** | 3.54 | **0.03** |
| Initial metastasis |  |  |  |  |
| M0 | 1 |  | 1 |  |
| M1 | 2.993 | **0.043** | 2.972 | **0.046** |
| Line of therapy |  |  |  |  |
| First | 1 |  |  |  |
| Second or more | 1.504 | 0.236 |  |  |
| Visceral metastasis |  |  |  |  |
| No | 1 |  |  |  |
| Yes | 3.457 | 0.063 |  |  |

The bold values are those that have *P* < 0.05 significance.

aberration, and local amplification of 1q21-q23 has been frequently observed in advanced metastatic cancers, unlike primary diseases tissue[15]. The telomeric amplification at 8p11, which includes eight genes (*ZNF703, PROSC, BRF2, RAB11FIP1, GOT1L1 ADRB3,* and *KCNU1*), was previously reported to be associated poor clinical outcomes in breast cancer[16,17]. The 8p11–12 amplicons have previously been shown to be associated with endocrine resistance and to include the histone methyltransferase WHSC1L1 and the receptor tyrosine kinase FGFR1. In our cohort, one of the 30 responders (3.3%) and two of the 15 non-responders (20%) have WHSC1L1 and FGFR1 duplication[18].

**Mutation signatures and clonal diversity.** Analysis of mutational signatures was performed to investigate whether mutational processes were responsible for genomic instability in non-responders[6]. We noted that mutational signatures related to the activities of the APOBEC family of enzymes (SBS 2, SBS 13), which are commonly found in breast cancer and are related to kataegis, contributed to 22.1% of the mutations in the non-responder group, compared to only 4.7% in the responder group[13]. Even in ER-positive patients, mutational signatures related to APOBEC were more predominant in non-responders than in responders (24.3% vs.4.7%) (Supplementary Fig. 4).

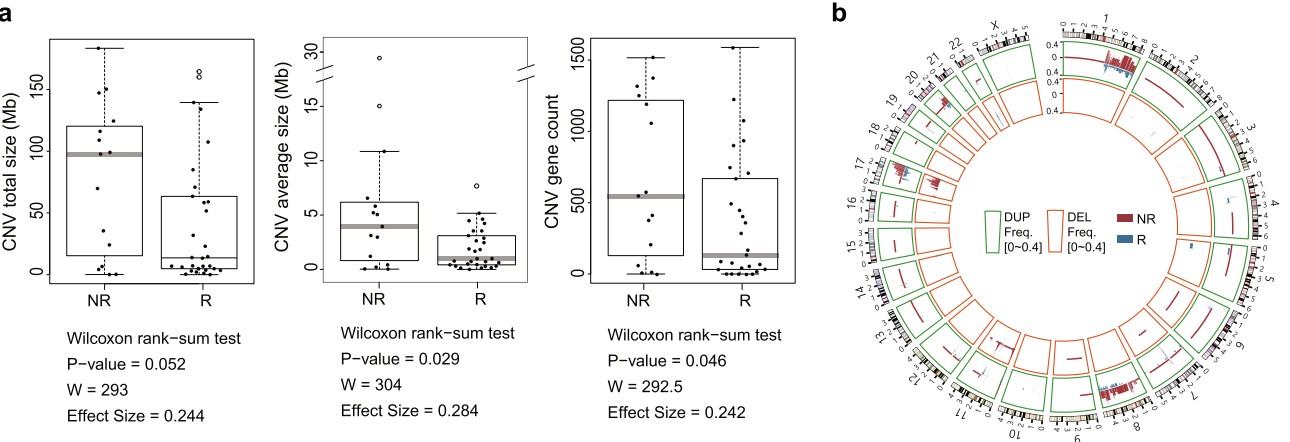

**Fig. 4 Comparison of copy number variants between responders and non-responders. a** Comparison of copy number variants (CNV, total and average length) and CNV gene count between responders and non-responders. *P* values were calculated by the Wilcox rank-sum test. **b** Circos plots depict CNV types and frequencies in the 30 responders and 15 non-responders. The frequency of duplications per genome base in each group is shown in histogram plots and indicated in the regions marked by green lines (max: 0.6). Also, the frequency of deletions (DEL) is indicated in the regions marked by orange lines (max: 0.8).

Recently, APOBEC signature has been linked to drug resistance in ER-positive breast cancer and ongoing tumor evolution by driving subclonal diversification[19]. In order to verify whether drug responses are associated with clonal diversity, we analyzed in non-responders and responders correlations between estimated clone counts and genetic features. The nonsynonymous mutational burden ($R = 0.52$ vs. 0.44) (Fig. 5b) and CNV total length ($R = 0.68$ vs. 0.36) (Fig. 5c), respectively, exhibited more positive correlations with clonal diversity in spite of a similar correlation between them ($R = 0.59$ vs. 0.53) (Fig. 5a) for non-responders than responders. The number of clones per patient was a median of 1 (mean = 2.03) in responders and a median of 2 (mean = 3.47) in non-responders ($P = 0.04$, one-side) (Fig. 5d). Patients with a higher number of clones tended to exhibit higher intratumor heterogeneity (ITH) (i.e., greater nonsynonymous mutation burden and larger CNV lengths). The proportion of patients with high intratumor heterogeneity was markedly greater in non-responders than in the responders (Fig. 5e).

## Discussion

In this study, we analyzed mutational and CNV burden in extreme responders to anti-ER or anti-HER2 therapy among metastatic breast cancer patients. In doing so, we discovered that responders harbored significantly fewer nonsynonymous mutations and a lower CNV burden, compared to non-responders. These findings confirm our hypothesis that extreme responses to anti-cancer therapy may be characterized by fewer genomic alterations.

Treatment responses are undergirded by various genomic factors. In line with this, we initially hypothesized that patients with larger numbers of genomic alterations would exhibit activation of alternative pathways that interfere with treatment responses. Supporting our hypothesis, preclinical and clinical data has suggested that resistance to therapy in ER-positive and HER2-positive patients constitutes complex molecular crosstalk between ER and HER2 pathways[20]. Meanwhile, co-targeting PI3K/mTOR pathways or the CDK4/6 complexes has been found to significantly delay resistance to endocrine therapy[21,22], and dual inhibition of the HER2-pathway has been shown to provide a more complete blockade of HER2 signaling[23]. Notwithstanding, these pivotal studies offer no insights into extreme responders to endocrine and HER2-targeting therapy. Accordingly, the

National Cancer Institute's Exceptional Responder Study (NCT02243592) is currently attempting to document large number of exceptional responders to understand their tumor biology and inform treatment decisions[24].

Ellis et al. performed massive parallel sequencing analysis to characterize responses to aromatase inhibitors[25]. Several pathways were enriched in the aromatase inhibitor-resistant subjects, including *TP53* signaling, DNA replication, and mismatch repair. The results were in accordance with ours, in that aromatase inhibitor-resistant individuals had larger numbers of point mutations and indels, genome-wide copy number alterations, and structural rearrangements. Chalmers et al. recently analyzed human cancer genomes and revealed that *TP53* gene mutations, which were also significantly enriched in our non-responder group, were associated with high tumor mutational burden[26].

Impaired p53 pathway signaling may impact cellular responses to DNA damage. A defective p53 pathway can lead to aberrant expression of p53 target genes, such as *PTEN*, *BRCA1*, and *RP1*, which directly affect both the recruitment of DNA repair proteins to sites of DNA damage and DNA damage repair potential[1]. Interestingly, however, genes involved in DNA mismatch repair, such as *MSH2, MSH6, MLH1*, and *PMS2*, were not enriched in our non-responder group.

Several studies have proposed that tumor mutational burden may be used as a predictive biomarker of immune checkpoint inhibitors[27,28]. Tumor-specific mutations created by DNA alterations have been found to result in the formation of novel proteins that allow the immune system to distinguish cancer cells from noncancer cells, and these neoantigens can enhance T cell reactivity[29]. Therefore, mutational load may be an indirect biomarker with which to predict tumor-specific T cell reactivity. Although our patients did not include those who were treated with immunotherapy, investigating whether non-responders with high mutational burden may be good candidates for treatment with immune checkpoint inhibitors would be interesting.

Tumors often evolve, and understanding a tumor's evolutionary trajectory may help to predict patient outcomes. In this study, genetic heterogeneity was more common in non-responders. Similarly, multiple studies have described associations between subclonal diversity and adverse clinical outcomes in cancers[30]. One previous study reported that intratumor heterogeneity in HER2 copy numbers was associated with shorter

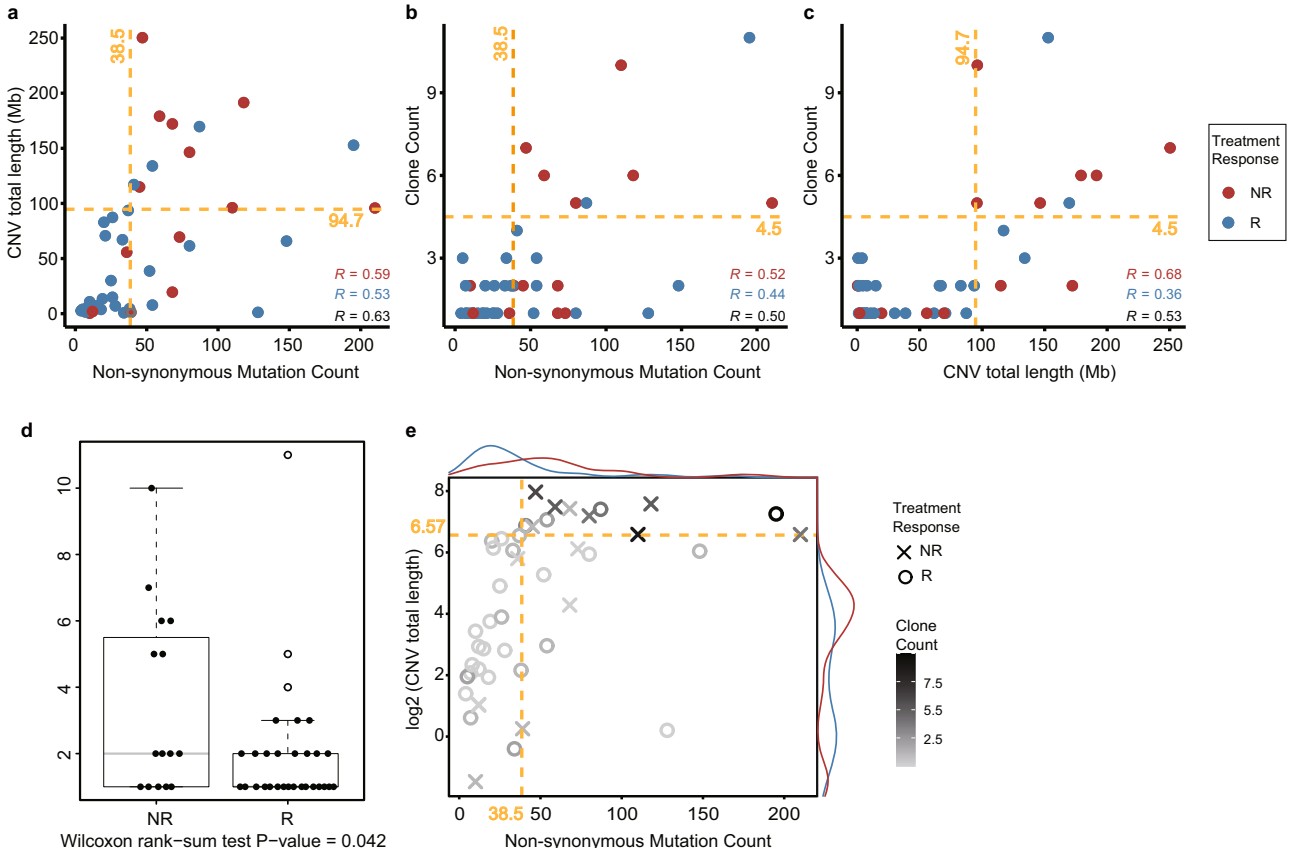

**Fig. 5 Comparison of CNVs, nonsynonymous mutations, and the number of clones. a** Correlation between total size of CNVs and nonsynonymous mutations. Correlation between number of clones and **b** nonsynonymous mutations and **c** total size of CNVs. **d** Comparison of the number of clones between responders and non-responders. **e** The number of clones increased in patients with both higher numbers of nonsynonymous mutations and larger CNVs. The color variation from black to light gray indicates higher to lower clone numbers, respectively. The suggested cut-off of each feature for distinguishing extreme responders from non-responders is marked with yellow line. The copy number burden has more discriminative power than various feature combinations owing to the high coefficient of variation (nonsynonymous mutations: 0.97, CNV: 1.11, clone: 0.94).

survival[31]. Likewise, the presence of subclonal diversity, as noted in the non-responder group, may reduce the therapeutic effect treatments in breast cancer patients.

This study has a few limitations that warrant consideration. One is that RNA sequencing was not feasible, although it would provide more accurate evaluation of CNVs. This was because FFPE blocks were old and RNA was degraded to a large degree. Also, the functional significance of each nonsynonymous variation and copy number variant was not validated. Lastly, our cohort comprised only a small number of HER2-positive breast cancer patients, which may show different mutational burdens. Nevertheless, the majority of our patients were ER-positive, and our analysis of ER-positive patients alone showed higher nonsynonymous mutational burden in non-responders. Lastly, all patients included in this analysis were Korean, which may hinder the generalizability of our findings to other ethnic groups. Despite these limitations, our study provides new insights into extraordinary responses to treatment in metastatic breast cancer patients.

In conclusion, extreme responders to treatment of metastatic breast cancer are characterized by low nonsynonymous mutational and low CNV burden.

## Methods

**Patient recruitment**. We prospectively and consecutively recruited primary breast cancer tissue samples and matched blood samples from patients who visited two institutions (Yonsei Cancer Center and Asan Medical Center, Korea) from April

2013 to February 2019. In addition, we also searched the metastatic breast cancer database at Yonsei Cancer Center ($n = 2548$). The criteria for extreme responders were (1) a complete or (2) partial response for more than two times the reported progression-free survival (PFS) for metastic breast cancer in historical data. The criteria for non-responders were (1) no shrinkage in tumor diameter and (2) progressive disease as the best response. Clinical information, including age, sex, treatment duration, best response to treatment, percent change in tumor size, previous treatment history, and survival data, were collected. Tumor response evaluation was conducted as per Response Evaluation Criteria in Solid Tumor (RECIST), version 1.1[32]. The study protocol was approved by the independent ethics committee and institutional review board of Severance Hospital and was conducted in accordance with the Declaration of Helsinki and Good Clinical Practice. All patients provided written informed consent for genomic testing in this study. Specimens were evaluated by a board-certified pathologist (J.S.K.) to identify tumor-bearing areas for DNA extraction.

### Genome variant analysis

*Whole exome sequencing and preprocessing.* Genomic DNA was isolated from formalin-fixed paraffin-embedded (FFPE) specimens using QIAamp DNA FFPE Tissue Kits (Qiagen). Genomic DNA was used for SureSelectXT Target Enrichment library generation (Agilent) and was then captured by Human All Exon V5 (Agilent). We performed whole exome sequencing analysis using Illumina HiSeq2500. For the improved quality of variant calls, we followed the Genome Analysis Toolkit (GATK) best practice of data pre-processing for variant discovery[33]. Sequencing reads for normal and tumor samples were aligned and processed to the human reference genome (UCSC hg19) using BWA-MEM v0.7.17 and Picard tools v2.19.0 (http://broadinstitute.github.io/picard/)[34].

*Single-nucleotide variants (SNVs) and indel calling.* Somatic SNVs and indels were identified in normal–tumor paired samples by Mutect2 in GATK v4.1.0.0, with the min-base-quality-score option set to 30[35]. SNVs were annotated and filtered with SnpEff and SnpSift v4.1 based on dbSNP v151[36–38]. Also, we annotated common

somatic mutations in the Catalog of Somatic Mutation in Cancer (COSMIC) database v86[39]. To minimize the calling of false-positive SNVs resulting from artifacts, such as FFPE contamination, we used the modules of GetPileupSummaries, CalculateContamination, CollectSequencingArtifactMetrics, and FilterByOrientationBias included in the GATK variant filtering.

To retain only high confidence nonsynonymous coding variants, we applied the following criteria for an additional filtration of the initial call set: (1) variants rejected by the Mutect2 filter, (2) variants included in noncoding regions, (3) variants with alternate allele counts <3, (4) variants with an allele frequency <0.1, and (5) variants with a total allele count (read depth) <20.

*Copy number variants (CNVs) calling.* CNVs were called using EXCAVATOR2 v1.1.2, with the option minimum mapping quality >30[40]. A paired mode was used to compare CNVs in tumor samples with matched controls. Genomics regions with an estimated copy number fraction >3 between tumor and control tissue were considered as duplications. Similarly, regions with fractions <1 were called as deletions. We considered a gene to be affected by CNVs if the entire exonic region of the gene was completely contained in the CNV calls. Genes with low coverage (average read-depth < 20) were removed to reduce false positivity. The entire process of genomic variant analysis is summarized in Supplementary Fig. 1.

*Analysis of mutational burden and group specificity.* One-tailed Wilcoxon rank-sum test was used to test whether mutational burden (nonsynonymous somatic mutations and CNVs) differed between the responder and non-responder groups. To test for group-specific enrichment of genomic variants, Fisher's exact test was conducted for each called variant (nonsynonymous mutations, indels, and CNVs), applying a cut-off $P$ value of 0.05. In Supplementary Fig. 5, we denoted genes with group-specific CNV selected by Fisher exact test in each group and genes with CNV that recurrence in both groups. The function impact of somatic SNVs was predicted using PROVEAN v1.1.5 or SIFT v6.2.1[41,42]. All statistical analyses were performed using R version 3.6 (http://www.r-project.org) with wilcox.test and fisher.test functions.

*Mutation signature and enrichment pathway analysis.* The relative contribution of COSMIC mutational signatures v3 was assessed within our responders and non-responders using Mutalisk with breast cancer-specific signatures based on PCAWG signature. And we performed cross-validatation using SigProfiler[43]. A matrix containing the information on the somatic mutation variants was created using the module of SigProfilerMatrixGenerator with exome option and then used for signature extraction and visualization through SigProfilerExtractor and SigProfilerPlotting. To proceed with the enrichment analysis, we selected genes that were mutated in more than two patients within each group. The selected gene list in each group was used as the input data for analyzing mutated signaling pathways in Enrchr[44].

*Inference of clonal populations and impurity.* The clonal populations of each patient with respect to CNVs and allelic counts were inferred using the PyClone with the Beta Binomial emission model[45]. PyClone was performed using binomial emission densities and the pior option of total copy number for higher confidence of clonal populations. The suggested cut-offs of each feature were determined using impurity for distinguishing extreme responders from non-responder. The Gini index impurity measure is one of the split criteria of the decision tree in the Classification. The smaller the Gini index means that the better the classification, and that Gini index used for cut-off value to distinguish the two groups. The Gini index was used to measure the impurity of mutational burden, copy number burden, and clones as follows: $\text{Gini} = 1 - \sum_{i=1}^{n} P_{ci}^2$, where $P_{ci}$ is the probability of class $C_i$[46].

**Reporting summary.** Further information on research design is available in the Nature Research Reporting Summary linked to this article.

## Data availability
We have deposited a dataset on the mutations and copy number variants obtained through analysis on the GitHub page (https://github.com/hellokeyworld/Extreme). All sequence data have been uploaded to NCBI under accession code PRJNA700464, with controlled access. The datasets generated during and/or analyzed during the current study are available from the corresponding author upon reasonable request. All the source data for Figs. 2–5 are available at Supplementary Data 1.

## Code availability
The custom code used during the current study are provide on the Zenodo page (https://doi.org/10.5281/zenodo.4457453)[47] and GitHub page (https://github.com/hellokeyworld/Extreme).

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

## Acknowledgements

This research was supported by a grant of the Korea Health Technology R&D Project through the Korea Health Industry Development Institute (KHIDI), funded by the Ministry of Health & Welfare, Republic of Korea (grant number: HI19C0430). This study was also supported by a faculty research grant from Yonsei University College of Medicine for 2014 (6-2014-0188). This work was supported by the National Research Foundation of Korea (NRF) grant funded by the Korea government (MSIT) (No. 2019R1A2C2008050). This study was supported by a faculty research grant of Yonsei University College of Medicine (6-2016-0081).

## Author contributions

S.M.L., E.K., J.S., and S.K. conceived the idea and S.M.L. and J.S. designed the study. S.M.L., K.H.J., S.I.K., S.P., H.S.P., B.W.P., Y.U.C., J.Y.K., G.M.K., J.H.Km., M.H.K., and J.S. contributed to obtaining patient and clinical information. J.S.K. performed immunohistochemical staining; and E.K., S.R.K., S.K., and N-.J.K. performed experimental analysis; and E.K. and S.K. conducted computational data analysis and generated figures. S.M.L. and E.K. wrote the manuscript. All authors supervised the project and edited the manuscript.

## Competing interests

The authors declare no competing interests.
