## [Peer Review File · Communications Biology]

Reviewers' comments:

Reviewer #1 (Remarks to the Author):

This manuscript presents a comparative genomic analysis between breast cancer patients showing exceptional response to anti-HER2 or hormonal therapy and non-responders. A higher mutational burden, higher copy number variation and increased overall intratumour heterogeneity are proposed as characteristic features of the non-responder group. It is noteworthy that these appear to differ significantly despite the small number of samples in each group. The question addressed in this study is an important one and the overall analyses and statistics employed seem correct. However, I do not believe this manuscript is publishable in its current form. I have the following major concerns:

1. While I appreciate exceptional responders to such therapies are rare, the cohorts available for comparison after filtering are small and this severely limits the power of identifying significant differences between the two groups. This is still, however, less of a concern than the fact that any findings resulting from this study should be validated in an external dataset. The authors should consider using data from resources like TCGA, ICGC or Hartwig Medical Foundation (all of which should have at least some samples with the right profile) both to supplement their discovery cohort as well as to validate their findings.
2. The authors mention the data were collected both from two centres in Korea as well as from a "metastatic breast cancer database". The authors should clarify how many samples were collected from each source and what exactly is the metastatic breast cancer database they are referring to (is it an internal or a public database? If it's the latter, provide a link to it).
3. There was a single TNBC patient included in the analysis. It may be worth excluding this sample as TNBC is quite a different subtype of cancer altogether and this would confer greater uniformity to the cohort.
4. The mutational signature analysis was performed using the 30 signatures which were reported in COSMIC in the previous version of their database. However, this has been recently updated to align with the results of the PCAWG consortium, resulting in an increased number of signatures (<https://cancer.sanger.ac.uk/cosmic/signatures/SBS/>). The authors should update their analysis to include all these signatures.
5. The authors use deconstructSigs and PyClone to investigate mutational signatures and clonality in the cohorts, respectively. The authors should confirm that their findings are robust by running other tools that serve the same purpose (e.g. SigProfiler for mutational signatures). In particular, I was surprised that the number of clones identified in a single bulk tumour sample can be as high as >60. The authors should run at least one other tool for clonal inference to verify their results.

6. The multivariate analysis model should also include the initial cancer stage, therapy given and (potentially) the age of the patient. E.g. it seems that around 73% of non-responders were initially diagnosed with stage 4 cancer compared to only 33% of patients showing good response to therapy.

7. Figure 4 and Extended data Fig.2B were missing.

8. The authors only investigate copy number variation in terms of length and number of segments. What about amplified or deleted genes? Are there any that differ between the two groups? It would also be beneficial to present the overall copy number variation by chromosome across all the samples in the two groups (e.g. in a heat map), to highlight if there are any recurrent patterns.

9. The authors highlight the increased mutational and copy number burden as features that could distinguish extreme responders from non-responders. Which of these two features would have more discriminative power (or should a combined score be considered)? What might be a good cut-off to use (a median mutational burden is suggested, but is this the one that has the best discriminative power)?

10. The authors examined progression free survival in their cohort in relation to response to therapy and mutational burden. What about overall survival?

11. There are a lot of spelling mistakes throughout the manuscript. This needs to be proofread by a native speaker. Additionally, some sentences could be rewritten to read a little clearer e.g. line 321.

12. The code employed in the analysis should be made freely available.

Minor comments:

- Within Table 1 there is a lack of consistency with using capital letters and the overall formatting could be improved.
- In supplementary figure 3, it would be good if some of the specific genes mentioned in the main text e.g. AF2, TTN, TP53, ATM, CENPF and MLLT4, were also shown in the figure.
- Line 228: the authors comment on the percentage of responders who received the listed therapy as their first line of treatment. It would be good to specify this also for the non-responders.
- Line 283: it is not entirely clear which of the two ways to assess CNV mutation burden the authors employed to categorize patients into two groups.
- The main parameters used to run GATK should be specified.

Reviewer #3 (Remarks to the Author):

The authors present an elegant analysis of the genomic landscape between responder and non-responder in metastatic breast cancer. The authors have gone to great length to precisely explain the rationale of the study, carried the relevant experiments and produced consonant findings thereof that are of relevance to cancer research and precision medicine application. Here the authors linked genomic alterations (copy number variations and point mutations) of breast tumours to the treatment responses of patient's afflicted by these tumours. It is a well-designed laboratory-based and computational study with following minor concerns.

Comment 1: line 165, the authors write that, "For minimizing false positive SNVs resulting from an Artifact such as FFPE contamination, using the module included in the process of GATK variant filtering". Did you mean to write that, "For minimizing false positive SNVs resulting from an Artifact such as FFPE contamination, WE USED the module included in the process of GATK variant filtering"?

Comment 2: line 167, the authors write that, "1) variants rejected variants by Mutect2 filter". Could the authors please delete the second "variants" in the text?

Comment 3: lines 241 and 248, the authors refer to Fig. 3A and Fig. 3B. However, Figure 3 does not have panels A and B. May the authors please correct this?

Comment 5: Can the Authors reframe the sentence "The relative contribution of 30 mutational signatures based on provided at COSMIC within responders and non-responders were identified using R package deconstructSigs (version 1.8.0)?"

Comment 6: Concerning the mutation burden, some of the responders had tumours with an equal or even higher mutational burden compared to the non-responders. For these, could the authors provide information about the mutated signalling pathway? Such information would help understand the biology underpinning of breast tumours that respond well to the applied treatments. Here the responders with high mutations burden could be compared to the non-responder with an equal or lower mutation burden. Also, could the authors examine the altered signalling pathways between the responders with tumours that have a high mutation burden vs the responders that have tumours with a lower mutation burden? Here, the Ma'ayan lab provides a web-based tool called Enchr that can make this task easy.

Comment 7: The authors write that "Comparison of mutation burden among ER-positive patients did not show a significant difference between responders and non-responders (45.96

vs. 258 66.21, $P=0.07$), although non-responders harbored more nonsynonymous mutations". Here also, could the authors provide some enrichment analyses to help explain the biological variations that impact treatment response?

Comment 8: On several instances, the Authors report statistical results by mean p-values only. Can the authors provide the other necessary measures, i.e., the test statistics (e.g., chi-square and rank-sum values) and the effect size (where relevant), together with the associated p-values?

Comment 9: There is a mixup with the figures. Firstly, the Authors have listed five figures in the figure legend, but the manuscript only has four figures. Also, there is a mention of "Circos plots visualizing copy number variants" in the figure 4 legend; however, there are no circos plots plot on figure 4. Also, extended figures are not provided. Could the Authors correct this?

Comment 10: Could the authors provide along with the manuscript the processed mutations and copy number datasets so other researchers may independently validate the findings reported in this paper?

Reviewers' comments

Reviewer #1 (Remarks to the Author):

This manuscript presents a comparative genomic analysis between breast cancer patients showing exceptional response to anti-HER2 or hormonal therapy and non-responders. A higher mutational burden, higher copy number variation and increased overall intratumour heterogeneity are proposed as characteristic features of the non-responder group. It is noteworthy that these appear to differ significantly despite the small number of samples in each group. The question addressed in this study is an important one and the overall analyses and statistics employed seem correct. However, I do not believe this manuscript is publishable in its current form. I have the following major concerns:

1. While I appreciate exceptional responders to such therapies are rare, the cohorts available for comparison after filtering are small and this severely limits the power of identifying significant differences between the two groups. This is still, however, less of a concern than the fact that any findings resulting from this study should be validated in an external dataset. The authors should consider using data from resources like TCGA, ICGC or Hartwig Medical Foundation (all of which should have at least some samples with the right profile) both to supplement their discovery cohort as well as to validate their findings.

Answer to Q1 - We have initially conducted a prospective analysis of extreme responders and non-responders among patients derived from metastatic breast cancer database during April 2013 to February 2019. In the initial analysis, we discovered that 18 extreme responders harbored a significant lower tumor mutation burden as compared to 8 non-responders. We then aimed to validate our findings in a separate cohort of extreme responders and non-responders in the external cohort (Asan Medical Center). Therefore, we additionally collected 29 patients (21 extreme responders and 8 non-responders) as a validation cohort. The validation cohort analysis revealed that although extreme responders harbored a lower TMB, it was not statistically significant (Extended Figure 2). Finally, we conducted a pooled analysis of 30 extreme responders and 16 non-responders in the final dataset to increase the statistical power of our study cohorts.

2. The authors mention the data were collected both from two centres in Korea as well as from a “metastatic breast cancer database”. The authors should clarify how many samples were collected from each source and what exactly is the metastatic breast cancer database they are referring to (is it an internal or a public database? If it's the latter, provide a link to it).

Answer to Q2 - We extracted the cases from the metastatic breast cancer database, which is the internal database in Yonsei Cancer Center. Therefore, this is not a public database, but rather a database originating from prospective collection of newly diagnosed metastatic breast cancer patients. We revised the manuscript in Page 6, line 96 that metastatic breast cancer database originates from one institution consisting of 2,548 patients.

3. There was a single TNBC patient included in the analysis. It may be worth excluding this sample as TNBC is quite a different subtype of cancer altogether and this would confer greater uniformity to the cohort.

Answer to Q3 - We agreed with the reviewer's suggestion and removed the TNBC patient in our study cohorts. We described the results of our re-analysis on 45 patients except for TNBC samples in the manuscript in Page 10. A comparative analysis of 15 responders and 30 non- responders indicated to confirm our hypothesis that extraordinary responders would have lower nonsynonymous mutation burden, CNV burden, and clonal diversity in metastatic breast cancer.

4. The mutational signature analysis was performed using the 30 signatures which were reported in COSMIC in the previous version of their database. However, this has been recently updated to align with the results of the PCAWG consortium, resulting in an increased number of signatures (<https://cancer.sanger.ac.uk/cosmic/signatures/SBS/>). The authors should update their analysis to include all these signatures.

Answer to Q4 - We appreciate the reviewer's advice, and we have updated the signatures up to version 3 for analysis of mutational signatures comparison of responders and non-responders (Extended Figure 4). All patients have BRCA1/BRCA2 germline mutation, so that the signatures 3 associated with BRCA1/BRCA2 deficiency is present both groups. In addition to signature 3, signatures 6 associated with mismatch repair deficiency, and signatures 1 relevant the age of diagnosis are commonly detected signatures in breast cancer. ¹ Signatures 2 and 13, which are related to APOBEC mutagenesis was more predominant in non-responders than responders (25.9% vs. 6.2%). The signature 9 appears to responders specific signatures (12.3%). According to previously reported that the signatures 9 was actually present in metastatic breast cancers but at relatively low levels. ² As a result of the mutation signature analysis among ER-positive patients, it was maintained that the results of the whole cohort as the same.

5. The authors use deconstructSigs and PyClone to investigate mutational signatures and clonality in the cohorts, respectively. The authors should confirm that their findings are robust by running other tools that serve the same purpose (e.g. SigProfiler for mutational signatures). In particular, I was surprised that the number of clones identified in a single bulk tumour sample can be as high as >60. The authors should run at least one other tool for clonal inference to verify their results.

Answer to Q5 - First of all, we do agree with the fact that it's important to confirm robust results by running other tools that serve the same purpose. In the case of Mutation Signature, the analysis results were confirmed to be identical using Mutalisk and SigProfiler. From the result of both tools, confirmed our argument that APOBEC associated signatures (2,3) predominate in non-responders over responders even though signatures differences existed (Figure R1).

Figure R1 The mutation signatures from SigProfiler and Mutalisk

A few methods have been developed to address ITH detection and quantification from NGS data in tumor samples, but it is difficult to define that ITH can be measured accurately with one WES sample per patient.³ According to the paper, some methods failed ITH detection and quantification (See table below).

Table 1. Main characteristics of ITH methods tested. The mean runtime is the mean time to process a TCGA sample. The success rate is the fraction of TCGA samples for which the method produced an output without error, with ASCAT calls as input only. The MATH score was computed in one step for all samples, using a table containing all mutations for all samples; the operation lasted 3.21s (std. 47.6 ms) for the protected dataset, and 3.39s (std. 11ms) for the public dataset. All time measurements were measured on a single cluster node with a 2.2 GHz processor and 3GB of RAM.

Method	CNA as input	Purity as input	Outputs tree (s)	Reference	Mean (std) runtime protected in seconds	Mean (std) runtime public in seconds	Success rate (protected)	Success rate (public)
MATH	no	no	no	[30]	<< 1	<< 1	100%	100%
EXPANDS	yes	no	no	[9]	891 (604)	267 (258)	89%	71%
PyClone	yes	yes	no	[6]	7,035 (8,464)	1,414 (1,415)	95%	99%
SciClone	yes	no	no	[7]	62 (48)	41 (51)	92%	78%
PhyloWGS	yes	yes	yes	[8]	13,258 (9,058)	4,730 (4,139)	95%	97%

Table from reference 3(J Abécassis et al.)

Clonality was also calculated using SciClone and EXPANDS in our study, but the detection failed, resulting in a missing value. Therefore, Pyclone, which can calculate the clonality of all patients, was used in the analysis. Additionally, clonality obtained by Pyclone is reported to have a higher correlation with B-SCITE than other measurement tools. B-SCITE is inferring tumor phylogenies and subclonal compositions from combined single-cell and bulk-sequencing data.⁴ Instead, we selected the clone consisted of clusters supported two or more SNVs to increase precision. As the results, the number of clones per patient was a median of 1 (Mean = 2.03) in responders and a median of 2 (Mean = 3.47) in non-responders (Wilcoxon rank-sum test $P = 0.04$, one-side).

6. The multivariate analysis model should also include the initial cancer stage, therapy given and (potentially) the age of the patient. E.g. it seems that around 73% of non-responders were initially diagnosed with stage 4 cancer compared to only 33% of patients showing good response to therapy.

Answer to Q6 - We appreciate reviewer's comments on the multivariate analysis. With regard to initial cancer stage, we categorized patients as either initially stage 4 or initially stage 1-3, and we represented as initial metastasis M1 (stage 4) or M0 (stage 1-3). With regard to age, we categorized patients as < 55 and more or equal to 55, which is the median value. Therapy is quite heterogeneous in each patient which includes both HER2-targeting agents, hormonal agents and chemotherapy. Therefore, we only categorized as line of therapy (first vs. second or more). Please see the revised Table 2.

7. Figure 4 and Extended data Fig.2B were missing.

Answer to Q7 - We apologize for the error. We have included Figure 5 and Extended data Fig. 2B in the final set.

8. The authors only investigate copy number variation in terms of length and number of segments. What about amplified or deleted genes? Are there any that differ between the two groups? It would also be beneficial to present the overall copy number variation by chromosome across all the samples in the two groups (e.g. in a heat map), to highlight if there are any recurrent patterns.

Answer to Q8 - As the reviewer suggested, we analyzed the gene with copy number variation. We calculated the number of genes with distinct CNVs per patient as defined CNV gene count. CNV gene count is a median of 134 (Mean = 358.7) in responders and a median of 594 (Mean = 660.9) in non-responders. The CNV gene count pass the statistical test in discriminating two groups (Wilcoxon rank-sum test $P = 0.046$, one-side). The spectrum of copy number variants for the entire patients was added in manuscript (Extended Figure 5). Also, we identified Region with copy number variation that is distinct between groups in the cytoband units. The Fisher test analysis shows that the CNV Regions that are specific to the NR group at 1q (1q21.2, 1q32.3, 1q41, 1q44) ,8q (8q11.22, 8q11.23, 8q12.3, 8q13.1, 8q13.2, 8q21.12, 8q24.13), and 17q25.2 are statistically significant (Fisher's exact test $P < 0.05$). The 8p11-12 amplicons have been previously associated with endocrine resistance and include the histone methyltransferase WHSC1L1, the receptor tyrosine kinase FGFR1. ⁵ We confirmed in our cohort that 1 out of 30 responders (3.3%), 2 out of 15 non-responders (20%) have WHSC1L1 and FGFR1 duplication.

Figure R2 The spectrum of copy number variants of cohort

9. The authors highlight the increased mutational and copy number burden as features that could distinguish extreme responders from non-responders. Which of these two features would have more discriminative power (or should a combined score be considered)? What might be a good cut-off to use (a median mutational burden is suggested, but is this the one that has the best discriminative power)?

Answer to Q9 – We appreciate the reviewer's suggestion. We present the cut-off for distinguishing extreme responders from non-responders by each feature using impurity. Gini index is measured impurity of mutation burden, copy number burden, and clone ; $Gini = 1 - \sum_{i=1}^n P_{ci}^2$, where P_{ci} is the probability of class Ci in a node. ⁶ The cut-off of nonsynonymous mutation count is 38.5, 94.7 for the CNV total length cut-off, and 4.5 for the clone cut-off. The cut-off of each feature is marked with a yellow dotted line in Figure 5. The copy number burden has more discriminative power than various feature combinations owing to the high coefficient of variation (Nonsynonymous mutations : 0.97, CNV :1.11, Clone :0.94)

Figure 5

Figure R3 Cut-off of each feature for distinguish extreme responders from non-responders marked with yellow line

10. The authors examined progression free survival in their cohort in relation to response to therapy and mutational burden. What about overall survival?

Answer to Q10 - The main objective of this study was to assess the genomic landscape of extreme responders to specific anti-cancer treatment as compared to non-responders. The extreme responders and non-responders were selected on the basis of progression-free survival and each treatment is described in Table 1. The OS data was not mature enough for analysis since none of the extreme responders had died at the time of data cutoff. In addition, we presumed that as OS was affected by the subsequent treatment that show heterogeneous responses, we did not include the OS analysis in the manuscript.

11. There are a lot of spelling mistakes throughout the manuscript. This needs to be proofread by a native speaker. Additionally, some sentences could be rewritten to read a little clearer e.g. line 321.

Answer to Q11 - Thank you for your valuable comments. The manuscript has been proofread by a native speaker. And regarding previous line 321, we rewrote the sentence to be more clear on the meaning.

12. The code employed in the analysis should be made freely available.

Answer to Q12 - We agreed with the reviewer's suggestion and provide the GitHub page that contains codes used in the analysis (<https://github.com/hellokeyworld/Extreme>).

Minor comments:

1. Within Table 1 there is a lack of consistency with using capital letters and the overall formatting could be improved.

Answer to q1 - We have revised Table 1 to ensure that all letters are consistent throughout.

2. In supplementary figure 3, it would be good if some of the specific genes mentioned in the main text e.g. AFF2, TTN, TP53, ATM, CENPF and MLLT4, were also shown in the figure.

Answer to q2 - Following the reviewer's advice, we added 6 genes added in the Extended Figure 3. The added 6 genes is 5 non-responder-specific genes (AFF2, TTN, TP53, ATM and MLLT4) and ERS1 associated with resistance to aromatase inhibitors (AI). We appreciate the careful note.

3. Line 228: the authors comment on the percentage of responders who received the listed therapy as their first line of treatment. It would be good to specify this also for the non-responders.

Answer to q3 - As the reviewer recommended, we added the proportion of patients who received the mentioned therapy as their first line of treatment. Please refer to page 11, line 191-193.

4. Line 283: it is not entirely clear which of the two ways to assess CNV mutation burden the authors employed to categorize patients into two groups.

Answer to q4 - We are sorry for causing any confusion. In previous Line 283, we divided into high CNV burden groups and low CNV burden groups based on the median value of CNV burden. We corrected the manuscript as the reviewer pointed out in page 13, line 233 to 235.

5. The main parameters used to run GATK should be specified.

Answer to q5 - We are sorry for the lack of a detailed description of the parameters to run GATK. More detailed information has been added to the Methods section of the manuscript. We refer to the GATK best practice workflow of somatic short variants and the GATK tutorial #11136 called '(How to) Call somatic mutations using GATK4 Mutect2' (<https://gatk.broadinstitute.org/hc/en-us/articles/360035889791--How-to-Call-somatic-mutations-using-GATK4-Mutect2-Deprecated->).

The main parameters of the run GATK are as follows.

- Mutect2
 - min-base-quality-score= 30
- FilterMutectCalls
 - min-median-base-quality = 30
 - min-median-mapping-quality = 30
 - normal-artifact-lod = 0.5
 - max-events-in-region = 50

Reviewer #3 (Remarks to the Author):

The authors present an elegant analysis of the genomic landscape between responder and non-responder in metastatic breast cancer. The authors have gone to great length to precisely explain the rationale of the study, carried the relevant experiments and produced consonant findings thereof that are of relevance to cancer research and precision medicine application. Here the authors linked genomic alterations (copy number variations and point mutations) of breast tumours to the treatment responses of patient's afflicted by these tumours. It is a well-designed laboratory-based and computational study with following minor concerns.

We truly appreciate reviewer's comments regarding the structure and the conduct of our study. To improve the scientific value of our results, we have corrected all the minor concerns suggested by the reviewer #3 as below.

Comment 1: line 165, the authors write that, "For minimizing false positive SNVs resulting from an Artifact such as FFPE contamination, using the module included in the process of GATK variant filtering". Did you mean to write that, "For minimizing false positive SNVs resulting from an Artifact such as FFPE contamination, WE USED the module included in the process of GATK variant filtering"?

Answer to Q1 - The reviewer is correct that we meant "For minimizing false positive SNVs resulting from an artifact such as FFPE contamination, we used the module of GetPileupSummaries, CalculateContamination, CollectSequencingArtifactMetrics, and FilterByOrientationBias included in the process of GATK variant filtering." Please see the revised sentence in page 7, line 124-127.

Comment 2: line 167, the authors write that, "1) variants rejected variants by Mutect2 filter". Could the authors please delete the second "variants" in the text?

Answer to Q2 - We thank the reviewer for correcting the typo. We removed the second "variants" in the text. Please see the revised sentence in page 7, line 129.

Comment 3: lines 241 and 248, the authors refer to Fig. 3A and Fig. 3B. However, Figure 3 does not have panels A and B. May the authors please correct this?

Answer to Q3 - As recommended by the reviewer, we labeled Figure 3 as A and B.

Comment 5: Can the Authors reframe the sentence "The relative contribution of 30 mutational signatures based on provided at COSMIC within responders and non-responders were identified using R package deconstructSigs (version 1.8.0)?"

Answer to Q5 - As the reviewer suggested, we reframed and updated the sentence as below. "The relative contribution of COSMIC mutational signatures v3 were assessed within our responders and non-responders using SigProfiler." Please refer to page 8, line 152-153.

Comment 6: Concerning the mutation burden, some of the responders had tumours with an equal or even higher mutational burden compared to the non-responders. For these, could the authors

provide information about the mutated signalling pathway? Such information would help understand the biology underpinning of breast tumours that respond well to the applied treatments. Here the responders with high mutations burden could be compared to the non-responder with an equal or lower mutation burden. Also, could the authors examine the altered signalling pathways between the responders with tumours that have a high mutation burden vs the responders that have tumours with a lower mutation burden? Here, the Ma'ayan lab provides a web-based tool called Enrchr that can make this task easy.

Answer to Q6 - We appreciate reviewer's advised on the mutated signalling pathway using Enrchr. Estrogen receptor signaling pathway and ERBB2 role in signal transduction and oncology associated with response of treatment is non-responders specific pathway (Table R1). The three genes that PIK3CA, AKT1, and ESR1 are associated with these two pathways and are previously known to induce drug resistance in breast cancer when mutations occur in these genes.⁷⁻⁹ In case, ESR1 mutations located in previously known hotspot regions, Y537S and D538G at the ligand-binding domain are occurred in non-responders. While there was one inframe deletion located hinge domain in responders, the functional impact is unpredictable and might be tolerated. The PH domain of AKT1 interacts with its kinase domain and maintains the protein in a closed and inactive state.¹⁰ We found that AKT1 E17K mutation located PH domain appears both groups, and AKT1 D323N mutation located in the kinase domain is only detected non-responders.

Groups	Pathway Name	Adjusted p-value (<0.01)
Non-Response	Tumor suppressor Arf inhibits ribosomal biogenesis	0.0004148
	Estrogen receptor signaling pathway	0.006786
	eIF4E and p70 S6 kinase regulation	0.007376
	ERBB2 role in signal transduction and oncology	0.007662
	Chronic myeloid leukemia	0.007812
Response	Hypoxia and p53 in the cardiovascular system	0.007867
	Termination of O-glycan biosynthesis	0.0000527
	O-linked glycosylation of mucins	0.002537
	Chronic myeloid leukemia	0.003513
	Endometrial cancer	0.009746

Table R1 Pathway comparison between Responders and Non-responders

The pathways of responders with high mutation burden are related protein modification. Otherwise, responders with low mutation burden's pathway is common in cancer (Table R2). The pathway associated with drug resistance could not be identified within the responders, regardless of the classification according to the mutation burden in responders.

Groups	Pathway Name	Adjusted p-value (<0.1)
Response with	Termination of O-glycan biosynthesis	0.009441
	O-linked glycosylation of mucins	0.06964

Groups	Pathway Name	Adjusted p-value (<0.1)
High TMB	Post-translational protein modification	0.07667
Response with LOW TMB	Chronic myeloid leukemia	0.06545
	Pathways in cancer	0.08258
	Phagosome	0.09972

Table R2 Pathway comparison between Responders with high TMB and Responders with low TMB

Groups	Pathway Name	Adjusted p-value (<0.05)
Response with High TMB	Termination of O-glycan biosynthesis	0.000993
	Tumor suppressor Arf inhibits ribosomal biogenesis	0.008493
	Angiogenesis	0.01067
	O-linked glycosylation of mucins	0.02185
	Post-translational protein modification	0.04721
	Estrogen receptor signaling pathway	0.04841
Response with LOW TMB	Chronic myeloid leukemia	0.001408
	Endometrial cancer	0.01074
	Pathways in cancer	0.01363
	Melanoma	0.02056

Table R3 Pathway comparison between high TMB and low TMB

Furthermore, we analysis of enrichment pathways between the patients have a high mutation burden vs low mutation burden (Table R3). As a result, the pathways related to tumor growth suppression, such as Tumor suppressor Arf inhibits ribosomal biogenesis, are common in non-responders and High TMB groups. ¹¹And estrogen receptor signaling pathway, which can indicate drug resistance, was observed only in the patients who have a high mutation burden.

Comment 7: The authors write that "Comparison of mutation burden among ER-positive patients did not show a significant difference between responders and non-responders (45.96 vs. 258 66.21, P=0.07), although non-responders harbored more nonsynonymous mutations". Here also, could the authors provide some enrichment analyses to help explain the biological variations that impact treatment response?

Answer to Q7 - Estrogen receptor signaling pathway, ERBB2 role in signal transduction and oncology, and Signaling by ERBB4 linked with the response of treatment are non-responders in ER positive patients specific pathway (Table R4). The three genes that PIK3CA, AKT1, and ESR1 are presented in all three pathways and PHLPP1 is involved in Signaling by ERBB4. The mutation of PHLPP1 is associated with mechanisms of resistance to anticancer drugs in melanoma via the loss of function for tumor suppressor gene and negative regulator of the AKT kinase. ^{12,13}

Groups	Pathway Name	Adjusted p-value (<0.01)
Non-Response in ER +	Tumor suppressor Arf inhibits ribosomal biogenesis	0.0002879
	Estrogen receptor signaling pathway	0.005178
	Chronic myeloid leukemia	0.005465
	ERBB2 role in signal transduction and oncology	0.005851
	Hypoxia and p53 in the cardiovascular system	0.006006
	Apoptotic signaling in response to DNA damage	0.007766
	Signaling by ERBB4	0.008882
	Integrated cancer pathway	0.009688
Response in ER +	Termination of O-glycan biosynthesis	0.001277

Table R4 Pathway comparison between Responders and Non-Responders in ER-positive subtype

Comment 8: On several instances, the Authors report statistical results by mean p-values only. Can the authors provide the other necessary measures, i.e., the test statistics (e.g., chi-square and rank-sum values) and the effect size (where relevant), together with the associated p-values?

Answer to Q8 - We appreciate reviewer's comments. We calculated the effect size and statistics for a compared feature between responders and non-responders. Furthermore, we newly added a discussion in the manuscript with the table R5 in Extended Table 1.

Feature	Median (R vs. NR)	Wilcoxon rank sum one-side test P-value	Wilcoxon rank sum statistics W	Effect size
Mutation burden	19.5 vs. 66	0.025	75	0.425
CNV Gene count	136.5 vs. 490	0.046	295.5	0.253
CNV Count	13 vs. 12	0.452	230.5	0.02
CNV Total Length (Mb)	13.60 vs. 97.67	0.052	293	0.244
CNV Average Length (Mb)	0.99 vs. 3.94	0.029	304	0.284
Clone count	1 vs. 2	0.042	293	0.260

Table 5 The Statistic and effect size for a compared feature

Comment 9: There is a mixup with the figures. Firstly, the Authors have listed five figures in the figure legend, but the manuscript only has four figures. Also, there is a mention of "Circos plots visualizing copy number variants" in the figure 4 legend; however, there are no circos plots plot on figure 4. Also, extended figures are not provided. Could the Authors correct this?

Answer to Q9 - We apologize for the errors. We have correctly replaced the figures and double-checked if all figures were in the right place. We assumed that errors have occurred during the transfer of the manuscript.

Comment 10: Could the authors provide along with the manuscript the processed mutations and copy number datasets so other researchers may independently validate the findings reported in this paper?

Answer to Q10 - We again appreciate the suggestion and provide a dataset of the mutation and copy number variants obtained through analysis on the GitHub page (<https://github.com/hellokeyworld/Extreme>).

References

- 1 Nik-Zainal, S. & Morganella, S. Mutational Signatures in Breast Cancer: The Problem at the DNA Level. *Clin Cancer Res* **23**, 2617-2629, doi:10.1158/1078-0432.CCR-16-2810 (2017).
- 2 Angus, L. *et al.* The genomic landscape of metastatic breast cancer highlights changes in mutation and signature frequencies. *Nat Genet* **51**, 1450-1458, doi:10.1038/s41588-019-0507-7 (2019).
- 3 Abecassis, J. *et al.* Assessing reliability of intra-tumor heterogeneity estimates from single sample whole exome sequencing data. *PLoS One* **14**, e0224143, doi:10.1371/journal.pone.0224143 (2019).
- 4 Malikic, S., Jahn, K., Kuipers, J., Sahinalp, S. C. & Beerenwinkel, N. Integrative inference of subclonal tumour evolution from single-cell and bulk sequencing data. *Nat Commun* **10**, 2750, doi:10.1038/s41467-019-10737-5 (2019).
- 5 Kwek, S. S. *et al.* Co-amplified genes at 8p12 and 11q13 in breast tumors cooperate with two major pathways in oncogenesis. *Oncogene* **28**, 1892-1903, doi:10.1038/onc.2009.34 (2009).
- 6 Alagiriswamy, P. S. a. S. Association Rule Based Similarity Measures for the Clustering of Gene Expression Data. *The Open Medical Informatics Journal*, 63-73, doi:10.2174/1874431101004010063 (2010).
- 7 Yang, S. X., Polley, E. & Lipkowitz, S. New insights on PI3K/AKT pathway alterations and clinical outcomes in breast cancer. *Cancer Treat Rev* **45**, 87-96, doi:10.1016/j.ctrv.2016.03.004 (2016).
- 8 Lopez-Cortes, A. *et al.* Mutational Analysis of Oncogenic AKT1 Gene Associated with Breast Cancer Risk in the High Altitude Ecuadorian Mestizo Population. *Biomed Res Int* **2018**, 7463832, doi:10.1155/2018/7463832 (2018).
- 9 Hanker, A. B., Sudhan, D. R. & Arteaga, C. L. Overcoming Endocrine Resistance in Breast Cancer. *Cancer Cell* **37**, 496-513, doi:10.1016/j.ccell.2020.03.009 (2020).
- 10 Caumanns, J. J. *et al.* Integrative Kinome Profiling Identifies mTORC1/2 Inhibition as Treatment Strategy in Ovarian Clear Cell Carcinoma. *Clin Cancer Res* **24**, 3928-3940, doi:10.1158/1078-0432.CCR-17-3060 (2018).
- 11 Golomb, L. & Oren, M. In the race for protection, ARF comes second. *Cell Death Differ* **20**, 1442-1443, doi:10.1038/cdd.2013.117 (2013).
- 12 Kalal, B. S., Upadhyay, D. & Pai, V. R. Chemotherapy Resistance Mechanisms in Advanced Skin Cancer. *Oncol Rev* **11**, 326, doi:10.4081/oncol.2017.326 (2017).
- 13 Suljagic, M. *et al.* Reduced expression of the tumor suppressor PHLPP1 enhances the antiapoptotic B-cell receptor signal in chronic lymphocytic leukemia B-cells. *Leukemia* **24**, 2063-2071, doi:10.1038/leu.2010.201 (2010).

Reviewers' comments:

Reviewer #1 (Remarks to the Author):

The authors have addressed some of my concerns in a satisfactory manner. However, a few of my questions have been rather superficially addressed and not awarded sufficient justification. In particular, there is still a lack of clarity on the following points:

Q1: I appreciate the authors' effort to validate their findings in a separate cohort of 29 cases, and the difficulty of finding suitable samples for this analysis. However, I must insist that additional validation in cohorts like TCGA and ICGC could still provide some further backing of the study (given that the authors' validation cohort didn't confirm the findings), despite the fact that these are primary and not metastatic tumours. The Hartwig Medical Foundation provides the perfect dataset for validation, but I appreciate gaining access to that dataset can be difficult, which is why I suggest looking at TCGA at least (and ICGC if the clinical data is suitable). From a quick investigation I can see there are 670 cases that received hormonal therapy and at least 40 who were treated with Herceptin (and possibly more) in TCGA.

Q5: It was good to see the comparative results on mutational signatures from SigProfiler and Mutalisk. I agree that the APOBEC signal seems to hold. How do the authors explain the fact that only SBS1 and SBS5 (both related to ageing) are found in responders by SigProfiler? Also, signature 3 is missing from the SigProfiler results despite being the most prevalent in Mutalisk. I would therefore argue the results are far from identical and perhaps the authors should explore multiple signature configurations by SigProfiler to check for such dissimilarities.

Q8: The authors misunderstood my point. I was suggesting that they should infer which specific cancer-related genes are amplified or deleted across the genome and produce an OncoPrint-type plot (similar to the one in Extended Fig 3) containing copy number variation in the top altered genes, comparing responders and non-responders (or this can be merged with Extended Fig 3). Of course there are also other ways to present that data, and I leave that to the authors' discretion. The authors highlighted a couple of genes that fell within copy number altered regions, but a comprehensive survey on all such genes is easily achievable and could yield useful insights.

Q9: I found it rather difficult to understand the method used for determining the cut-off. Could the authors explain it in more detail? What is meant by "impurity"? They should also add this explanation to the methods section, since this refers to Fig 5 of the manuscript.

Q12: I appreciate the fact that the authors' have deposited some bash scripts illustrating their procedures for mutation, copy number calling and clonality analysis. It would be potentially even more useful to include the code that was used to analyse the resulting data (i.e. the code that was used to compare the two groups, responders and non-responders, and to generate the figures/tables).

Reviewer #3 (Remarks to the Author):

The authors have convincingly addressed all the concerns that I had with the initial submission.

They have now reported all statistical test results to include the p-values, test-statistics and effect sizes. Furthermore, as suggested, they have deposited online the datasets and source code used to arrive at the conclusions and claims made in the paper so other researchers could reproduce the results.

Reviewers' comments:

Reviewer #1 (Remarks to the Author):

The authors have addressed some of my concerns in a satisfactory manner. However, a few of my questions have been rather superficially addressed and not awarded sufficient justification. In particular, there is still a lack of clarity on the following points:

Q1: I appreciate the authors' effort to validate their findings in a separate cohort of 29 cases, and the difficulty of finding suitable samples for this analysis. However, I must insist that additional validation in cohorts like TCGA and ICGC could still provide some further backing of the study (given that the authors' validation cohort didn't confirm the findings), despite the fact that these are primary and not metastatic tumours. The Hartwig Medical Foundation provides the perfect dataset for validation, but I appreciate gaining access to that dataset can be difficult, which is why I suggest looking at TCGA at least (and ICGC if the clinical data is suitable). From a quick investigation I can see there are 670 cases that received hormonal therapy and at least 40 who were treated with Herceptin (and possibly more) in TCGA.

Answer to Q1 – As suggested by the reviewer, we used the TCGA breast cancer database to analyze the tumor mutation burden in association with survival. Tumor mutation burden was calculated based on the primary tumors *via* whole exome sequencing of 506 patients, excluding duplicate cases among the 670 cases. We only selected patients who received adjuvant hormonal therapy or Herceptin.

A previous paper reported that a high tumor mutation burden is found in 5% of all breast cancers and is more common in metastatic tumors and primary breast cancer tumor mutation burden has a lower value than metastasis breast cancer.¹ The median tumor mutation burden of our 30 metastatic cohorts was 36 (average 49.58) and the median tumor mutation burden of the primary TCGA cohort was 25 (average 40.54), confirming that it was comparable to the reported results.

For a total of 506 samples, we divided 252 samples into the TMB-High group and 254 samples into the TMB-Low group based on the median nonsynonymous mutation value of 25 to confirm our finding in this dataset.

Although early breast cancer harbored fewer genomic alterations than metastatic breast cancer, it was confirmed that the survival was poor when the tumor mutation burden was high. We noted a significant OS difference between TMB-high group and TMB-low group (p-value 0.035, Figure R1).

Figure R4. Survival analysis in TCGA cohort grouped by tumor mutation burden (p-value 0.035)

Q5: It was good to see the comparative results on mutational signatures from SigProfiler and Mutalisk. I agree that the APOBEC signal seems to hold. How do the authors explain the fact that only SBS1 and SBS5 (both related to ageing) are found in responders by SigProfiler? Also, signature 3 is missing from the SigProfiler results despite being the most prevalent in Mutalisk. I would therefore argue the results are far from identical and perhaps the authors should explore multiple signature configurations by SigProfiler to check for such dissimilarities.

For a detailed explanation, the answer is described by dividing the question in Q5 into Q5-1 and Q5-2.

Q5-1) How do the authors explain the fact that only SBS1 and SBS5 (both related to ageing) are found in responders by SigProfiler?

Answer to Q5-1 – We are grateful to reconsider the analysis results thanks to the reviewer's advice. We analyzed the mutation signatures of COSMIC version 3 using an option for exonic data provided by the latest version of SigProfiler and found SBS13 along with SBS1, SBS5 in responders. The results and method parts of the *Mutation signatures and clonal diversity* were modified based on the updated analysis results.

Additionally, APOBEC associated signatures, SBS2 and SBS13, are not represented in all breast cancers but are reported under special conditions. The APOBEC activity is the most common process in hyper-mutated tumors and the APOBEC associated signatures predominate in non-responders over responders.

Figure R5 Results of mutation signatures analysis with COSMIC version 3 obtained from SigProfiler

Q5-2) Also, signature 3 is missing from the SigProfiler results despite being the most prevalent in Mutalisk.

Answer to Q5-2 – The number of signatures used in the mutation signature analysis is the first cause of the difference. Mutalisk provides cancer-specific signatures, even though it does not include prevalent signatures such as SBS5 and SBS13. Figure R3 shows the result of analyzing the types of mutation signatures of COSMIC version 3 and their proportion for various cancer types. The portion of mutation signatures in breast cancer was listed in order, SBS5 (97.27%), SBS1 (91.80%), SBS2 (82.73%), SBS13 (78.8%), SBS3 (39.01%), and the remaining signatures have a portion of less than 30%.² So we recalculated with the Mutalisk tool considering breast cancer-specific signatures based on Figure R3 and confirmed that SBS3 is extracted in Mutalisk.

In the case of SigProfiler, the number of minimum signatures can be fixed, but the stability core is more important, signatures attributed can be decomposed with fewer signatures. According to previous reports, SBS3, known to be associated with the process of defective homologous recombination-based DNA repair, was reconstructed into SBS8 or SBS5 from the multiple myeloma dataset.³ The SBS3 is not necessary to explain the patterns of SNV mutations in the dataset and SBS8 and SBS5 emerged as the

most significant processes and the ones that are likely active. Also, signatures attributed to the same underlying mutagenic processes may correlate within and between individuals. The SBS3, InDel signatures ID6 and ID8, and rearrangement signatures SV3 and SV5 indicate different aspects of defects in homologous recombination-mediated repair. Even if SBS3 is not extracted from our cohort, ID8 associated with the deficient homologous recombination-based DNA repair has been detected.

Figure R6. The number of SBS mutations attributed to each mutational signature for each cancer type.

Q8: The authors misunderstood my point. I was suggesting that they should infer which specific cancer-related genes are amplified or deleted across the genome and produce an OncoPrint-type plot (similar to the one in Extended Fig 3) containing copy number variation in the top altered genes, comparing responders and non-responders (or this can be merged with Extended Fig 3). Of course there are also other ways to present that data, and I leave that to the authors' discretion. The authors highlighted a couple of genes that fell within copy number altered regions, but a comprehensive survey on all such genes is easily achievable and could yield useful insights.

Answer to Q8 – We appreciated reviewer's comments on the copy number variation analysis. We extracted a gene with group specific and frequently occurring CNV. As shown in the figure below, we denoted genes with group-specific CNV selected by fisher exact test in each group and genes with CNV that recurrence in both groups. This Figure of oncoplot of CNV replaces the existing Extended Figure5. More details are described in section named *Analysis of mutational burden and group specificity* in the method part of the manuscript.

Extended Figure 5

Figure R7 The Onco-plot of copy number variants of cohort

Q9: I found it rather difficult to understand the method used for determining the cut-off. Could the authors explain it in more detail? What is meant by “impurity”? They should also add this explanation to the methods section, since this refers to Fig 5 of the manuscript.

Answer to Q9 – The Gini index impurity measure is one of the split criteria of the decision tree in the Classification. Decision tree proceeds in a way that increases the purity of each node and reduces impurity or uncertainty after classification. The smaller the Gini index means that the better the classification, and that Gini index used for cut-off value to distinguish the two groups.

For example, if a data set was split in two ways, A and B, let's determine which of the A and B method better distinguished the data set. (Figure R5)

Figure R8 Example of Gini index

The Gini index when you apply the classification method of A is 0.497. The Gini index for method B is 0.3305.

We already described impurity in the section named *Inference of clonal populations and impurity*. (line166, reference 21) To help readers understand, we supplemented the method section with more detailed explanations.

Q12: I appreciate the fact that the authors’ have deposited some bash scripts illustrating their procedures for mutation, copy number calling and clonality analysis. It would be potentially even more useful to include the code that was used to analyse the resulting data (i.e. the code that was used to compare the two groups, responders and non-responders, and to generate the figures/tables).

Answer to Q12 - As the reviewer suggested, the codes used in the resulting data were additionally uploaded to Github.

Reviewer #3 (Remarks to the Author):

The authors have convincingly addressed all the concerns that I had with the initial submission. They have now reported all statistical test results to include the p-values, test-statistics and effect sizes. Furthermore, as suggested, they have deposited online the datasets and source code used to arrive at the conclusions and claims made in the paper so other researchers could reproduce the result

<Reference>

- 1 Karn, T. *et al.* Tumor mutational burden and immune infiltration as independent predictors of response to neoadjuvant immune checkpoint inhibition in early TNBC in GeparNuevo. *Ann Oncol* **31**, 1216-1222, doi:10.1016/j.annonc.2020.05.015 (2020).
- 2 Alexandrov, L. B. *et al.* The repertoire of mutational signatures in human cancer. *Nature* **578**, 94-101, doi:10.1038/s41586-020-1943-3 (2020).
- 3 Maura, F. *et al.* A practical guide for mutational signature analysis in hematological malignancies. *Nat Commun* **10**, 2969, doi:10.1038/s41467-019-11037-8 (2019).

REVIEWERS' COMMENTS:

Reviewer #1 (Remarks to the Author):

The authors have addressed all my queries in a satisfactory manner. I recommend acceptance of the manuscript.